# Lactate, an Essential Metabolic Marker in the Diagnosis and Management of Pediatric Conditions

**DOI:** 10.3390/diagnostics15070816

**Published:** 2025-03-23

**Authors:** Alina Belu, Nina Filip, Laura Mihaela Trandafir, Elena Lia Spoială, Elena Țarcă, Diana Zamosteanu, Gabriela Ghiga, Jana Bernic, Alina Jehac, Elena Cojocaru

**Affiliations:** 1Department of Morphofunctional Sciences I—Pathology, Faculty of Medicine, “Grigore T. Popa” University of Medicine and Pharmacy, 700115 Iasi, Romania; belu_alina@d.umfiasi.ro (A.B.); zamosteanu.diana@d.umfiasi.ro (D.Z.); elena2.cojocaru@umfiasi.ro (E.C.); 2Department of Morphofunctional Sciences II—Biochemistry, Faculty of Medicine, “Grigore T. Popa” University of Medicine and Pharmacy, 700115 Iasi, Romania; nina.zamosteanu@umfiasi.ro; 3Department of Mother and Child—Pediatrics, Faculty of Medicine, “Grigore T. Popa” University of Medicine and Pharmacy, 700115 Iasi, Romania; laura.trandafir@umfiasi.ro (L.M.T.); gabriela.ghiga@umfiasi.ro (G.G.); 4Department of Surgery II—Pediatric and Orthopedic Surgery, Faculty of Medicine, “Grigore T. Popa” University of Medicine and Pharmacy, 700115 Iasi, Romania; tarca.elena@umfiasi.ro; 5Discipline of Pediatric Surgery, “Nicolae Testemițanu” State University of Medicine and Pharmacy, MD-2001 Chisinau, Moldova; jana.bernic@usmf.md; 6Second Dental Medicine Department, Faculty of Dental Medicine, “Grigore T. Popa” University of Medicine and Pharmacy, 700115 Iasi, Romania; alina.jehac@umfiasi.ro

**Keywords:** lactate, pediatric disease, metabolic status, anaerobic metabolism, predicting clinical outcomes

## Abstract

Measurement of circulating lactate is an essential diagnostic tool in pediatric medicine, playing a crucial role in assessing metabolic status and tissue oxygenation. Initially regarded as a byproduct of anaerobic metabolism, recent research has expanded our understanding of lactate’s roles across various physiological systems, from energy metabolism to immune modulation and neurological health. Elevated lactate levels are widely utilized to monitor critical conditions such as sepsis, trauma, and hypoxic–ischemic injury, offering valuable prognostic information in intensive care settings. Notably, lactate dynamics—particularly trends in serial measurements—are more effective than single readings for predicting clinical outcomes, especially in sepsis and trauma. Measurement of circulating lactate in different body fluids (blood, cerebrospinal fluid, and umbilical blood) provides critical insights into neonatal health and central nervous system involvement. However, challenges remain, including the need for non-invasive and rapid point-of-care testing, particularly in neonatal populations. Our aim was to review and synthesize the current literature on the role and particularities of measurement of circulating lactate in pediatric pathology. Emerging technologies, such as machine learning models and small molecule inhibitors, show promise in advancing lactate regulation and predicting hemodynamic instability. As the role of lactate in pediatric pathology continues to evolve, optimizing measurement protocols and exploring new therapeutic strategies will enhance early detection, intervention, and clinical outcomes for critically ill children.

## 1. Introduction

Lactate plays a critical role in pediatric pathology as both a biomarker and a metabolic product, offering valuable insights into various acute and chronic conditions [1]. The earliest documented observation of lactic acid as a pathological finding can be attributed to Scherer’s 1843 case reports, which also represent the first recognition of hyperlactacidemia in the context of septic and hemorrhagic shock [2]. More than a century later, Huckabee’s pivotal research conclusively demonstrated that lactic acidosis often occurs in the context of severe illness and that tissue hypoperfusion plays a central role in its pathophysiology [3].

However, although previously regarded as a byproduct of metabolism, lactate is now acknowledged as a critical component in cellular energy production and a key regulator of various physiological processes [4]. Lactate is a multifunctional molecule that is synthesized in substantial amounts during metabolic activities, encompassing both aerobic and anaerobic pathways [5]. While often seen as a waste product linked to hypoxia [6], lactate is actually a vital energy source, continuously produced and utilized by cells in significant amounts under fully aerobic conditions [5]. Recent research reveals that lactate also plays a significant role in cellular signaling and may impact gene transcription through epigenetic mechanisms [7], functioning as a multifunctional signaling molecule with important anti-inflammatory, antioxidant, and immunomodulatory properties [5].

In pediatric populations, the clinical relevance of measuring lactate levels is profound [8,9]. Elevated lactate concentrations often indicate tissue hypoperfusion and metabolic distress, particularly in critical situations like septic shock [10]. Evidence suggests that lactate clearance may serve as a more reliable predictor of outcomes compared to initial lactate levels, with significant correlations found between lactate clearance and survival rates in pediatric sepsis [11]. Moreover, high lactate levels are linked to increased mortality in children suffering from septic shock, highlighting the importance of lactate as a metabolic marker in clinical settings [12]. The significance of lactate extends beyond its role in metabolism, providing essential insights into physiological responses across various clinical scenarios, particularly those related to hypoxia and shock.

This narrative review seeks to examine the multifaceted roles and implications of lactate within the pediatric population, considering its significance in metabolic processes, cellular signaling, and potential therapeutic applications.

## 2. Hyperlactatemia—Definitions and Classifications

Blood lactate concentration is determined by the balance between its production and clearance [13]. The definition of elevated lactate levels is not universally standardized. However, in healthy children, normal venous lactate levels typically range from 0.5 to 2.2 mmol/L, while arterial lactate levels typically range from 0.5 to 1.6 mmol/L (mmol/L = mg/dL × 0.11) [14]. For lactate measured in capillary blood, age-specific reference values are as follows: 1.1–3.5 mmol/L for infants 0–2 months old, 1–3.3 mmol/L for children 3–24 months old, and 1–2.4 mmol/L for children older than 2 years [14]. Critically ill patients with an admission lactate concentration exceeding 2.5 mmol/L should be closely monitored for signs of clinical deterioration. Additionally, individuals with lactate levels below this threshold should still be considered for serial lactate measurements to enable early detection of potential worsening [15].

In clinical practice, it is essential to differentiate between hyperlactatemia and lactic acidosis, as these represent distinct clinical conditions. Whereas hyperlactatemia is characterized by a moderate elevation in blood lactate levels (ranging from 2 to 4 mmol/L) without accompanying metabolic acidosis, lactic acidosis is characterized by significantly elevated lactate concentrations, typically exceeding 5 mmol/L, along with metabolic acidosis [16]. Thus, hyperlactatemia is not invariably associated with metabolic acidosis, as lactate itself is not directly responsible for acidosis. The primary cause of acidosis in conditions such as hyperlactatemia, commonly seen in sepsis, is the accumulation of protons (H+) in the body, particularly from the hydrolysis of ATP during glycolysis. In cases of insufficient oxygen or mitochondrial dysfunction, the body is unable to utilize these protons effectively, resulting in a drop in pH [17]. Lactic acid, possessing a pKa of 3.86, exists predominantly in its ionized form at physiological pH, analogous to pyruvic acid [18]. Furthermore, the conversion of pyruvate to lactate does not result in the production of protons; rather, this reaction consumes two protons, which contradicts the anticipated outcome [5]. During glycolysis, two protons are generated in the form of NADH + H+ and two pyruvate molecules [17]. These redox equivalents are subsequently utilized in either the mitochondrial electron transport chain (under aerobic conditions) or in the reduction of pyruvate to lactate (under anaerobic conditions) [17]. Consequently, although glycolysis is associated with localized acidification, the subsequent metabolic processes do not contribute to a net proton accumulation. Thus, although lactate is formed and utilized continuously by our cells, even under fully aerobic conditions, in large quantities, lactic acidosis always indicates a severe metabolic disturbance, often due to tissue hypoperfusion, metabolic dysregulation, toxins, or congenital metabolic disorders [19]. For instance, in sepsis-related hyperlactatemia, acidosis arises from coupled proton and lactate transmembrane transport during glycolysis [20].

Over time, several classification systems have been developed to better categorize and explain the various factors that contribute to elevated lactate levels in the body. In 1976, Cohen and Woods proposed a classification of hyperlactatemia into two distinct types: type A, which is associated with insufficient oxygen delivery to tissues, and type B, which occurs in the absence of tissue hypoxia. The majority of clinical cases of lactic acidosis, both in children and adults, do not stem from hypoxia: within type B hyperlactatemia, three subtypes were identified: B1, linked to impaired lactate clearance such as in hepatic insufficiency; B2, associated with drugs or toxins; and B3, connected to inborn metabolic disorders [21,22].

An alternative classification of hyperlactatemia, based on the underlying mechanisms of lactate accumulation, was recently proposed by Muller et al. [17]. Elevated lactate levels can arise under three primary conditions: (a) increased pyruvate production, which may result from factors such as non-specific stimulation of glycolysis by respiratory alkalosis, catecholamines, or beta-agonists; metabolic disturbances associated with malignancies; or alterations in cellular redox potential, as seen in alcohol consumption or diabetic ketoacidosis; (b) decreased pyruvate utilization, as seen in hypoxia-related hyperlactatemia (such as “type A” hyperlactatemia, caused by inadequate oxygen delivery or tissue perfusion), pyruvate dehydrogenase dysfunction, commonly associated with thiamine deficiency, or mitochondrial dysfunction, whether inherited or acquired; and (c) decreased lactate clearance, primarily observed in severe hepatic failure [17].

## 3. Lactate—Friend or Foe? Classic and Current Theories

It is a common misconception that lactates are merely a harmful byproduct of metabolism resulting from low oxygen levels or insufficient blood flow [23]. In reality, lactate serves multiple roles, with its function extending beyond this simplified view. Emerging evidence indicates that hyperlactatemia may be useful in specific clinical conditions, such as traumatic brain injury and severe hypoglycemia [24]. Table 1 provides a comparison of the classic and current conceptions of lactate’s functions across various physiological systems, highlighting its evolving roles in metabolism, immunity, respiratory function, cardiovascular health, muscle activity, neurological processes, and bone health. This underscores the expanding understanding of lactate beyond its traditional association with anaerobic metabolism, offering new insights into its potential therapeutic applications in multiple clinical contexts.

## 4. The Role of Lactate in Pediatric Pathology

Over the last decades, it has become increasingly acknowledged that trauma patients experience the “blood vicious triad”, consisting of acidosis, hypothermia, and coagulopathy [39]. As a result, there has been an enhanced focus on evaluating local or systemic indicators of metabolic acidosis to gauge the severity of injury, assess the effectiveness of treatments, and offer prognostic insights [40,41]. Several biochemical markers signal acidosis, with lactate levels being one of the easiest and most commonly used serum indicators [9,17]. Lactate is now frequently regarded as a surrogate marker for injury severity [42], in comparison to other acidosis markers such as base deficit [43]. Ramanathan et al. prospectively evaluated the predictive value of admission serum lactate in pediatric trauma, finding that elevated lactate levels correlated with higher Injury Severity Scores (ISSs), increased need for interventions, and worse outcomes [44]. While lactate levels above 4.7 mmol/L were found to be highly specific for severe injury, lactate below 2.0 mmol/L was reassuring in ruling out significant trauma; however, lactate levels between 2.0 and 4.7 mmol/L showed limited predictive value for injury or outcomes [44]. Furthermore, lactate is an important prognostic indicator in trauma patients, as demonstrated by Zhang et al., who found significantly lower blood lactate levels in the good prognosis group (1.74 ± 0.42) compared to the poor prognosis group (2.52 ± 0.36) (*p* < 0.001) [7]. Moreover, Colak et al., in a study encompassing 154 pediatric patients with multi-trauma, reported that elevated blood lactate concentrations are strongly correlated with the severity of clinical outcomes, particularly the necessity for inotropic support: the median lactate level in patients requiring inotropic therapy was 5.3 mmol/L, significantly higher than the 1.3 mmol/L observed in those not requiring such intervention (*p* < 0.01) [45]. This differential highlights the prognostic significance of lactate, with increased lactate concentrations associated with extended durations of invasive mechanical ventilation (IMV), inotropic drug administration, and prolonged pediatric intensive care unit (PICU) stays. These findings reinforce lactate’s potential as a valuable biomarker for predicting and managing clinical outcomes in pediatric trauma cases [45].

In sepsis, lactate serves as a key biomarker for assessing tissue hypoperfusion and impaired cellular metabolism, reflecting the severity of the condition [46]. However, in the pediatric population, studies are limited, and their findings are often inconsistent or even conflicting. For instance, while Hatherill et al. reported no significant relationship between lactate levels and in-hospital mortality in pediatric intensive care unit patients [47], Kim et al. have demonstrated that elevated lactate can serve as an early indicator of mortality in children with sepsis and septic shock [48]. Also, the optimal timing for lactate measurement in septic patients within the intensive care unit remains a subject of ongoing debate. Early lactate measurement may serve as a significant predictor of mortality [49]. More recent studies advocate for serial lactate measurements to provide a more reliable assessment of clinical prognosis. Abdelaziz et al. emphasizes the importance of monitoring lactate level trends over time as a prognostic indicator for mortality in pediatric intensive care unit patients with severe sepsis or septic shock [50]. Increased lactate levels and impaired lactate clearance six hours post-admission were linked to an elevated risk of mortality [50]. The findings of this study are consistent with the results of Moustafa et al. who examined lactate levels in 76 critically ill children at the time of admission and again 6 h later. Their analysis revealed that initial lactate levels demonstrated limited sensitivity and specificity in predicting mortality (AUC 0.519, *p* = 0.789). While lactate levels at admission were found to be ineffective in predicting mortality, lactate clearance after 6 h emerged as a significant predictor of poor outcomes, with an AUC of 0.766 (*p* < 0.001), highlighting the greater prognostic value of lactate clearance over initial lactate concentrations in critically ill patients [51].

In surgical conditions, such as intussusception, lactate levels can serve as an important biomarker for assessing the severity of the condition, with elevated levels potentially indicating ischemia of the intestines or tissue hypoxia. Monitoring lactate may aid in predicting patient outcomes and guide timely decision-making regarding medication, surgical treatment interventions and antibiotic therapy [52,53]. Lee et al. evaluated 249 pediatric patients with intussusception, identifying significant differences in blood gas parameters between patients with unfavorable and favorable outcomes, particularly in pH, lactate, and bicarbonate levels. Multivariable regression analysis revealed that both pH and lactate were independently associated with unfavorable outcomes, with higher lactate levels correlating to increased positive predictive values for adverse outcomes, particularly at thresholds of ≥2.5 mmol/L and ≥3.0 mmol/L [52]. Although lactate shows potential as a biomarker for outcome prediction, in an analysis involving 39 patients with suspected intussusception who underwent abdominal ultrasound screening, Tamas et al. reported no significant difference in lactate levels between children with suspected (1.7 ± 0.7 mmol/L) and confirmed intussusception (1.9 ± 1.1 mmol/L) [54]. These findings suggest that lactate levels do not serve as a reliable diagnostic marker for this surgical condition. Among pediatric patients undergoing cardiac surgery, Loomba et al. found that serum lactate levels at the time of admission are a significant predictor of inpatient mortality risk. Specifically, those who experienced mortality had higher lactate levels (5.5 vs. 4.1 mmol/L), with a standardized mean difference of 1.80 (95% CI: 0.05–3.56, *p* = 0.04) [55].

Lactic acid metabolism, particularly in the context of disrupted gluconeogenesis, anaerobic glycolysis, and alterations in acid–base homeostasis, plays a critical role in the pathophysiology of various metabolic disorders [5]. In children, elevated lactate levels, especially in the absence of hypoxia, may indicate underlying disorders, such as inborn errors of metabolism, including mitochondrial diseases or pyruvate dehydrogenase deficiency, where the capacity for aerobic metabolism is impaired [16].

Lactic acidosis is a primary cause of metabolic acidosis, and in most cases, the differential diagnosis of hyperlactatemia is initiated in the context of metabolic acidosis [19]. Figure 1 presents a concise classification of metabolic acidosis based on anion gap levels, with an emphasis on hyperlactatemia.

Additional insights can be obtained by measuring plasma lactate levels in both the fed and fasted states. This approach is especially helpful when examining glycogen storage disorders (GSDs). Lactic acidosis in the fasted state is a significant indicator of GSD type I (glucose-6-phosphatase deficiency), whereas in GSD type III (amylo-1,6-glucosidase deficiency), lactate levels tend to rise most noticeably after eating [56]. When diagnosing these conditions, it is crucial to collect samples both before and after meals to ensure lactic acidosis is not overlooked.

## 5. The Assessment of Lactate Levels—Particularities in Children

Measurement of circulating lactate is a pivotal diagnostic tool in pediatric medicine, providing valuable insights into the metabolic status and tissue oxygenation of patients [55]. Blood lactate remains the most commonly used measurement, but cerebrospinal fluid [57] and umbilical blood lactate levels [58] provide critical diagnostic information in specific clinical contexts. In clinical practice, lactate levels are typically measured using either venous or arterial blood samples, although point-of-care testing methods have become increasingly common for faster results [59]. The most accurate representation of systemic lactate levels, particularly in cases involving shock, sepsis, or severe tissue hypoxia, is provided by arterial blood lactate measurements [59]. However, obtaining arterial blood in pediatric patients, particularly in neonates and infants, presents significant challenges due to their smaller anatomical structures and the invasive nature of the procedure. Furthermore, arterial lines or punctures are necessary, both of which carry risks such as bleeding, infection, and vascular injury [60]. In this context, the most commonly employed technique in pediatric clinical practice is venous blood lactate testing due to its relative ease of collection and reduced invasiveness compared to arterial blood sampling [61]. While venous lactate concentrations generally correlate well with systemic lactate levels, the presence of shock or poor perfusion may cause discrepancies between arterial and venous values [62]. Despite these limitations, venous lactate measurement is particularly useful in emergency departments and intensive care settings where rapid assessment is required [61]. Table 2 summarizes the key studies evaluating both arterial and venous lactate levels in pediatric populations.

In neonates, the ability of the foetus to clear lactate is limited, making umbilical blood lactate a sensitive marker of oxygen deprivation during labor [66]. The clearance of lactate, such as through the Cori cycle, requires energy for gluconeogenesis in the liver. This process is supported by oxygen-dependent ATP production in the mitochondria, which facilitates the conversion of lactate back into glucose. Therefore, when oxygen supply is compromised, lactate clearance becomes less efficient, and lactate accumulation occurs more rapidly [67]. Elevated umbilical arterial lactate levels were associated with an increased risk of neonatal encephalopathy and other adverse perinatal outcomes [68]. In a prospective longitudinal cohort study encompassing 160 pregnant women in the active phase of labor at term, a critical umbilical artery lactate concentration > 9.1 mmol/L accurately predicted Apgar scores < 7 at 5 min and the need for neonatal unit admission, with sensitivities of 76.47% and 61.90%, and specificities of 91.55% and 91.30%, respectively [69]. Additionally, umbilical artery lactate levels > 11.2 mmol/L were highly sensitive (100%) for predicting hypoxic–ischemic encephalopathy development in neonates, with a specificity of 88.39% [69]. These findings align with the conclusions of a systematic review encompassing 12 original studies [70], which highlighted the strong predictive value of umbilical lactate in assessing neonatal outcomes, particularly in relation to acidosis and neurological injury. Significant correlations have been observed between umbilical lactate levels and key indicators such as pH, base excess and 5 min Apgar scores, reinforcing its role as a reliable biomarker for evaluating neonatal health [70]. Furthermore, the high specificity (93%) and moderate sensitivity (69.7%) of umbilical lactate in predicting conditions such as hypoxic–ischemic encephalopathy underscore its potential as a cost-effective and accessible tool for the early assessment and management of neonatal complications [70].

Capillary lactate measurements may be utilized in point-of-care settings and for non-invasive testing, particularly in neonates and young children. However, although capillary blood samples, obtained via heel stick or finger stick, provide an easy and rapid method for lactate assessment [71], peripheral vasoconstriction—a common occurrence in critically ill or septic children—can result in inaccurate lactate readings, limiting the utility of this method in severe illness [72]. However, research in pediatric populations reveals inconsistent findings and varying recommendations. For instance, Walther et al. reported that venous samples are preferable to capillary samples when arterial samples cannot be obtained [72]. On the other hand, a strong correlation was found between capillary and arterial lactate concentrations (r = 0.98, *p* < 0.001) in a study on 25 newborns, suggesting capillary lactate measurements are comparable to arterial ones in neonates [73]. In children, the potential utility of capillary lactate measurement is promising in pre-hospital settings, particularly in distinguishing neurological conditions, such as epileptic seizures, from febrile seizures, syncope, and psychogenic nonepileptic seizures [74]. Capillary lactate levels were significantly higher in patients with epileptic seizures compared to those with febrile seizures (*p* < 0.0007) or syncope (*p* < 0.0204), with an ROC curve area of 0.71 (95% CI 0.61–0.80); at a cutoff of >3.9 mmol/L, pre-hospital capillary lactate demonstrated 49% sensitivity and 92% specificity for distinguishing seizure type [74].

In addition to arterial, venous, and capillary lactate measurements, lactate levels in cerebrospinal fluid (CSF) are particularly useful in pediatric patients with suspected central nervous system (CNS) involvement, such as meningitis, encephalitis, or hypoxic–ischemic injury, with elevated CSF lactate indicating hypoxia or infection within the CNS [75]. However, obtaining CSF via lumbar puncture can be more technically challenging in neonates and infants, even for the most experienced physicians, due to smaller anatomical spaces and risks of infection, spinal hematoma, and later onset of epidermoid tumors or cerebral herniation [76]. Elevated CSF lactate concentrations are often indicative of impaired cerebral metabolism, and they can help differentiate bacterial meningitis from viral or aseptic forms of meningitis [77]. A study involving 216 children aged 1 month to 15 years who presented to the emergency department with meningitis (60 diagnosed with bacterial meningitis and 156 with viral meningitis) demonstrated that CSF lactate levels exhibit high sensitivity and specificity in differentiating between the two etiologies: a CSF lactate threshold of 3 mmol/L was found to be highly indicative of bacterial meningitis, while levels in viral meningitis typically remained below 2 mmol/L [78]. Besides infections involving the CNS, CSF lactate shows promise in the early identification of children with mitochondrial diseases. Yamada et al. demonstrated that CSF lactate concentrations exceeding 2.2 mmol/L mg/dl offer a highly reliable diagnostic marker for mitochondrial diseases affecting the central nervous system, such as Leigh encephalomyelopathy and MELAS (mitochondrial encephalomyopathy, lactic acidosis, and stroke-like episodes), with a sensitivity of 94.1% and specificity of 100% [79]. Additionally, when blood lactate levels fell within the 2.2–4.4 mmol/L range, a lactate CSF/blood ratio greater than 0.91 provided a valuable diagnostic tool, demonstrating perfect sensitivity and specificity [79]. Moreover, Magner et al. demonstrated that brief seizures lasting less than 2 min do not elevate CSF lactate levels, suggesting that CSF lactate can reliably differentiate mitochondrial disorders with seizures from epilepsy, as long as the sample is not obtained shortly after prolonged seizures, a crucial consideration given that many mitochondrial patients present with seizures [80]. However, healthcare professionals should be aware that CSF lactate may show slight variations with age. For infants up to 24 months, average 90th percentile values of 1.8 mmol/L may be applicable, while in older children, age-adjusted reference intervals should be considered, particularly when values approach the 90th percentile [81].

The measurement of urinary lactate levels, while considered non-invasive, has shown limited promise as a biomarker in clinical settings. Studies indicate that the mean urinary lactate concentrations in children with Autism Spectrum Disorder (ASD) do not significantly differ from those in control populations. Additionally, no significant associations have been found between urinary lactate levels—whether low or high—and clinical features of mitochondrial dysfunction, such as respiratory, auditory, or visual impairments [82].

Thus, the measurement of circulating lactate serves as a critical diagnostic method, providing invaluable insights into metabolic status and tissue oxygenation across diverse clinical settings. Its versatility as a marker across various fluids, coupled with its potential to guide early management, makes lactate indispensable in pediatric clinical practice.

## 6. Future Directions of Research

In the era of technological advancements, Sughimoto et al. hypothesized that machine learning models applied to arterial waveforms and perioperative characteristics can predict blood lactate levels in pediatric ICU patients [82]. Their machine learning algorithm had the potential to non-invasively and continuously predict blood lactate levels, which serve as a marker of hemodynamic instability, offering an accurate tool that could prove clinically valuable [83]. Thus, the ability to non-invasively and continuously predict blood lactate levels using machine learning models represents a promising advancement, offering a real-time tool for detecting hemodynamic instability and potentially improving patient outcomes. However, further independent validation in other populations is needed before these models can be fully trusted [83,84].

An interesting path for future research in lactate regulation would be the exploration of small molecule inhibitors targeting key enzymes such as hexokinase-2 (HK-2) [85] and lactate dehydrogenase A (LDHA) [86], as well as monocarboxylate transporters (MCT1 and MCT4) [87], which play crucial roles in lactate production and transport. Investigating the modulation of these proteins could open new avenues for developing therapeutic strategies aimed at regulating lactate levels and mitigating disease progression [88].

## 7. Conclusions

Measurement of circulating lactate is a crucial tool in pediatric medicine, offering valuable insights into the metabolic status of children across a variety of acute and chronic conditions. Sequential lactate measurements, in particular, have proven to be more effective than single readings for predicting outcomes, especially in pediatric sepsis. A comprehensive understanding of lactate dynamics is essential for early detection, timely intervention, and improved clinical outcomes.

Recent research has broadened our perspective on lactate, revealing its dual role not just as a byproduct of glycolysis, but also as a key regulator of cellular function and metabolism. However, challenges remain in pediatric practice, particularly regarding the inconsistent correlation between venous and arterial lactate levels. These disparities highlight the need for continued investigation to optimize lactate measurement protocols.

In summary, advancing our knowledge of lactate’s role in pediatric health is vital for enhancing diagnostic accuracy and therapeutic strategies, ultimately improving outcomes for children in critical care settings.

## Figures and Tables

**Figure 1 diagnostics-15-00816-f001:**
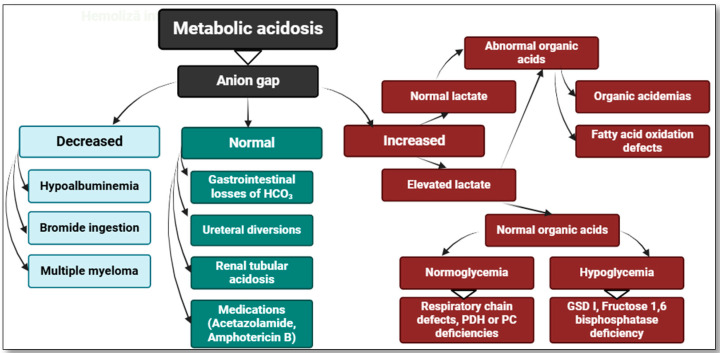
Differential diagnosis of metabolic acidosis—focus on hyperlactatemia. PDH—pyruvate dehydrogenase, PC—pyruvate carboxylase, GSD I—glycogen storage disease type I. Created with BioRender.com (accessed on 12 December 2024).

**Table 1 diagnostics-15-00816-t001:** Shifting perspectives on lactate: from byproduct to key mediator across multiple physiological systems.

Lactate		
Function	Classic Conceptions	Current Conceptions
Metabolism	Byproduct of anaerobic metabolism linked to muscle fatigue, especially during high-intensity activity [25]	Energy substrate, as well as main gluconeogenic precursor [26]
Immunity	Not traditionally seen as having a significant impact on immune responses.	Suppression of T cell activity in tumor microenvironment [27], CD8+ T cell-dependent tumor growth inhibition in mice bearing MC38 tumors following subcutaneous administration of sodium lactate [28]
Respiratory	Exercise-induced lactic acidosis stimulates ventilation and triggers hyperventilation at RCP (Respiratory Compensation Point) [29].	Lactate produced by alveolar type II cells suppresses inflammatory alveolar macrophages and mitigates acute lung injury by shifting macrophage cytokine expression toward an anti-inflammatory phenotype [30]
Cardiovascular	Elevated lactate levels within the first 12 h following successful resuscitation from cardiac arrest, reflecting hypoxia and associated anaerobic metabolism, were linked to higher mortality in children [31].	Half-molar lactate infusions may serve as a potential treatment for fluid resuscitation, demonstrating enhanced cardiac function, as indicated by increases in cardiac output, stroke volume, and left ventricular ejection fraction in healthy individuals (when compared to sodium-matched hypertonic sodium chloride) [32].
Muscular	Serum lactate was identified as a potential biomarker for muscle fatigue following intense exercise [33].	Lactate serves as an energy source for muscles, enhancing exercise capacity and endurance, while also supporting muscle recovery and adaptation following exercise [34].
Neurologic	Lactate production in the brain was thought to result from insufficient oxygen supply, impaired oxidative metabolism, or a mismatch between glycolysis and oxidative processes [35].	Lactate is now recognized as an essential fuel for the brain, playing a pivotal role in supporting cognitive function and neuronal activity, particularly in processes such as learning and memory, cerebral blood flow, neurogenesis, cerebral microangiogenesis, energy metabolism, and neuroprotection [36,37].
Bone health	No established connection to bone health.	Lactate may mediate the bone anabolic effects during high-intensity interval training by inducing osteoblast differentiation, offering a potential cost-effective therapeutic strategy for bone augmentation [38].

**Table 2 diagnostics-15-00816-t002:** Main studies on arterial and venous lactate levels in pediatric populations.

Study, Publication Year, Region	Design	Period	Population Sample	Disease	Department	Main Findings
Murdoch et al., 1994, London [63]	Comparative study	-	7 children, 2.3–10.8 years old, mean age 6.5 years old	Septic shock (*n* = 3), ARDS * (*n* = 2) and severe pulmonary hypertension (*n* = 2)	PICU **	The mean difference between arterial and mixed venous lactate was 0.02 mmol/L, with limits of agreement from −0.20 to 0.24. The differences were clinically insignificant (*p* = 0.36).
Fernández Sarmiento et al., 2016, Colombia [64]	Retrospective study	3 years and 4 months (January 2009–May 2012)	42 children, 1 month–17 years and 11 months old, mean age 2.3 years old	Sepsis and/or septic shock	PICU	A strong correlation between arterial and central venous lactate levels was found (Spearman’s rho = 0.897, *p* < 0.001). No significant differences were found between arterial and central venous lactate regarding age, weight, or diagnosis.
Samaraweera et al., 2017, London [62]	Retrospective study	3 years and 4 months (June 2012–October 2015)	60 children ≤17 years old, mean age 4.4 (SD = ±4.4) years old	Sepsis	PICU	A venous lactate concentration ≤2 mmol/L may serve as a surrogate for arterial lactate during the initial management of pediatric sepsis. If the venous lactate level exceeds 2 mmol/L, confirmation with an arterial sample is required.
Phumeetham et al., 2017, Bangkok, Thailand [65]	Prospective study	1 year (October 2013–October 2014)	48 children, 1 month–18 years old, median age 4.5 years old	Shock (54.1% septic, 12.5% neurogenic, 16.7% hypovolemic, 16.7% cardiogenic)	PICU	Venous and arterial lactate showed strong correlation (r = 0.962, *p* < 0.0001). The mean difference was 0.20 mmol/L (95% CI: 0.08–0.32), with limits of agreement from −0.74 to 1.13 mmol/L. Venous lactate is a reliable alternative to arterial sampling in shock.

ARDS * Acute respiratory distress syndrome; PICU ** Paediatric Intensive Care Unit.

## Data Availability

The data presented in this study are available on request from the corresponding author.

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
