# Peer review of "Lactate, an Essential Metabolic Marker in the Diagnosis and Management of Pediatric Conditions"

_diagnostics, 2025, doi:10.3390/diagnostics15070816_

Round 1
Reviewer 1 Report
Comments and Suggestions for Authors
General: This review is clearly written and covers a relevant clinical topic. Overall the message is consistent and well-referenced and important for clinicians. I have one (standard nowadays) question and a couple of moderate/minor comments which I assume are not very difficult to address.
Standard Question: no part of this manuscript was written with AI ?
Minor/moderate comments:
General: Some references are not appropriately placed. Please re-check them.
In the Abstract:
Please change "Lactate measurement" ->> "Measurement of circulating lactate ..
In the introduction:
L48: "Several decades" are in fact more than a century .....
L92 The sentences here convey a confusing message. Especially since some references also indirectly allude to Robergs who incorrectly claimed/claims that glycolysis does not produce lactic acid.
The simple stoichiometry 1 glucose -> 2 lactate + 2 protons, remains valid.
Also each lactate is exported together with a proton to the extracellular space. And this is also observed in (patho)physiology (see Brooks/ Gladden / Rabinowitz). The body just has a range of buffering systems to handle the protons, which causes the [lactate] and pH to often follow different trajectories.
So please modify your current sentence:
"Thus, hyperlactatemia is not invariably associated with acidosis as the commonly held belief regarding the systemic accumulation of lactic acid is not entirely accurate."
Likewise soften or remove:
"Consequently, although glycolysis is associated with a 101 localized acidification, the subsequent metabolic processes do not contribute to a net 102 proton accumulation"
Also the sentence that hyperlactatemia as such without acidosis is innocent, is incorrect. Hyperlactemia - also without acidosis - is a very worrying and prognostically bad sign, e.g. liver failure.
"Thus, hyperlactatemia can occur with normal tissue perfusion and 103 oxygenation, without significant metabolic disruption, whereas lactic acidosis indicates a severe metabolic disturbance, often due to tissue hypoperfusion, metabolic dysregulation, toxins, or congenital metabolic disorders."
Please modify the above sentence as well.
In discussing the types of lactic acidosis (LA), please stress that most clinical LA, both in kids and adults LA does not result from hypoxia.
P4, Table 1:
"Lactate may serve as a potential treatment for fluid resuscitation,.. "
it is not entirely clear what is meant here. Please clarify
In general plasma lactate is the more modern and correct term than serum lactate.
L 158 Please nuance "reliable" into "important"
L171 change "hypoxia" into "hypoperfusion"
L195 be more specific, ischemia of what ?
L203 please change "lactic acid" into "lactate", because that is what's measured
L206 id.
L208 id.
L271 please explain here or earlier that clearance of lactate (e.g. Cori cycle) essentialy requires oxygen.
L326 and elsewhere please convert mg/dL to mmol/L for consistency.
L361 please soften this overconfident prediction. ML algorithms require independent validation in other populations before they can be trusted.
L362 This paragraph undercuts what you (correctly) said earlier, namely that lactate is signal of severe disease and not a mediator.
L369 This paragraph with interesting but clinically yet unproven treatments also undercuts the message of a review on plasma lactate as a diagnostic tool.
Also refs 88,89 do not seem relevant in this context.
So in this paper that covers a different subject, it can be left out.
Better to stress that the first therapeutic focus in the context of lactate measurements as discussed in this paper is to act upon it in conventional ways.
E.g. HD-management, respiratory management, transfusion, antibiotics, etc.
Or just leave this therapeutic future paragraph entirely out for clarity.
Author Response
Comment 1: General: This review is clearly written and covers a relevant clinical topic. Overall the message is consistent and well-referenced and important for clinicians. I have one (standard nowadays) question and a couple of moderate/minor comments which I assume are not very difficult to address.
Standard Question: no part of this manuscript was written with AI ?
Response 1: No part of the manuscript has been written with AI. All content was written by our research team.
Minor/moderate comments:
Comment 2: General: Some references are not appropriately placed. Please re-check them.
Response 2: Thank you for pointing that out. We have re-checked the references and ensured they are appropriately placed in the manuscript.
Comment 3: In the Abstract:
Please change "Lactate measurement" ->> "Measurement of circulating lactate ..
Response 3: Thank you for the suggestion. We have changed "Lactate measurement" to "Measurement of circulating lactate" in both the abstract and the body text.
Comment 4: In the introduction:
L48: "Several decades" are in fact more than a century .....
Response 4: Indeed. We have modified accordingly.
Comment 5: L92 The sentences here convey a confusing message. Especially since some references also indirectly allude to Robergs who incorrectly claimed/claims that glycolysis does not produce lactic acid.
The simple stoichiometry 1 glucose -> 2 lactate + 2 protons, remains valid.
Also each lactate is exported together with a proton to the extracellular space. And this is also observed in (patho)physiology (see Brooks/ Gladden / Rabinowitz). The body just has a range of buffering systems to handle the protons, which causes the [lactate] and pH to often follow different trajectories.
So please modify your current sentence:
"Thus, hyperlactatemia is not invariably associated with acidosis as the commonly held belief regarding the systemic accumulation of lactic acid is not entirely accurate."
Response 6: Thank you for your comment. We have revised the sentence for clarity. Please see the lines 97-101.
Comment 7: Likewise soften or remove:
"Consequently, although glycolysis is associated with a 101 localized acidification, the subsequent metabolic processes do not contribute to a net 102 proton accumulation"
Response 7: We agree. For better clarity, we have removed the sentence.
Comment 8: Also the sentence that hyperlactatemia as such without acidosis is innocent, is incorrect. Hyperlactemia - also without acidosis - is a very worrying and prognostically bad sign, e.g. liver failure.
"Thus, hyperlactatemia can occur with normal tissue perfusion and 103 oxygenation, without significant metabolic disruption, whereas lactic acidosis indicates a severe metabolic disturbance, often due to tissue hypoperfusion, metabolic dysregulation, toxins, or congenital metabolic disorders."
Please modify the above sentence as well.
Response 8: Thank you for your comment. We have modified the phrase accordingly.
Comment 9: In discussing the types of lactic acidosis (LA), please stress that most clinical LA, both in kids and adults LA does not result from hypoxia.
Response 9: Thank you for highlighting that important point. We have included this valuable information in lines 123-124.
Comment 10: P4, Table 1:
"Lactate may serve as a potential treatment for fluid resuscitation,.. "
it is not entirely clear what is meant here. Please clarify
Response 10: Indeed, the initial formulation was unclear. We hope it is clearer now.
Comment 11: In general plasma lactate is the more modern and correct term than serum lactate.
Response 11: : Indeed, but for accuracy, we retained the formulation used in the original sources.
Comment 12: L 158 Please nuance "reliable" into "important"
Response 12: Ok.
Comment 13: L171 change "hypoxia" into "hypoperfusion"
Response 13: Ok. Done.
Comment 14: L195 be more specific, ischemia of what ?
Response 14: Thank you for your suggestion. We have revised the phrase accordingly.
Comment 15: L203 please change "lactic acid" into "lactate", because that is what's measured. L206 id. L208 id.
Response 15: Done.
Comment 16: L271 please explain here or earlier that clearance of lactate (e.g. Cori cycle) essentialy requires oxygen.
Response 16: Thank you for your suggestion. We included more details in lines 284-289.
Comment 17: L326 and elsewhere please convert mg/dL to mmol/L for consistency.
Response 17: Done.
Comment 18: L361 please soften this overconfident prediction. ML algorithms require independent validation in other populations before they can be trusted.
Response 18: We agree. We revised the lines
Comment 19: L362 This paragraph undercuts what you (correctly) said earlier, namely that lactate is signal of severe disease and not a mediator.
Response 19: We agree that lactate is primarily a signal of severe disease rather than a mediator. However, it’s important to clarify that what was discussed in this paragrash refers to future directions of research. The proposal to explore small molecule inhibitors targeting key enzymes like hexokinase-2 (HK-2), lactate dehydrogenase A (LDHA), and monocarboxylate transporters (MCT1 and MCT4) represents an emerging area of investigation. This could eventually lead to a better understanding of how modulating lactate metabolism may offer potential therapeutic strategies for regulating lactate levels and influencing disease progression. So, while the paragraph suggests an exciting future avenue, it doesn't negate the current understanding but rather points to potential future shifts in how we might view lactate's role in disease.
Comment 20: L369 This paragraph with interesting but clinically yet unproven treatments also undercuts the message of a review on plasma lactate as a diagnostic tool.
Also refs 88,89 do not seem relevant in this context.
So in this paper that covers a different subject, it can be left out.
Better to stress that the first therapeutic focus in the context of lactate measurements as discussed in this paper is to act upon it in conventional ways. E.g. HD-management, respiratory management, transfusion, antibiotics, etc.
Or just leave this therapeutic future paragraph entirely out for clarity.
Response 20: We agree. We have removed the entire paragraph, as well as references 88 and 89, for clarity.

Reviewer 2 Report
Comments and Suggestions for Authors
Lactate plays an important role in the management of critically ill patients in pediatric emergency and intensive care medicine.
Under normal conditions, lactate metabolism produces a small amount of ATP. However, by accelerating the glucose metabolism process, significantly more ATP can be produced. Therefore, a clinical condition associated with increased glucose metabolism may lead to elevated lactate levels, as the capacity of the Krebs cycle is limited. Many conditions and treatments in critically ill patients lead to an increase in glucose metabolism. One example is the increased sympathetic nervous system activation, which is markedly present in shock conditions. In such cases, lactate can serve as an exchangeable fuel between tissues (liver, kidneys, muscles) and even between cells (astrocytes, neurons) through lactate shuttles.
Unlike the oxygen-dependent Cori cycle (hepatic and renal gluconeogenesis), lactate metabolism allows for rapid energy production, and even exogenous lactate can be utilized as a fuel in this process. As a result, the traditional concept of lactate as an indicator of tissue hypoxia in shock conditions is being questioned. This relationship can be more problematic than suggested in guidelines, particularly in sepsis, where lactate clearance is impaired.
This review written on lactate may shed light on the management of critically ill children and may support the trend in the areas of research it suggests.
Author Response
Comment 1: Lactate plays an important role in the management of critically ill patients in pediatric emergency and intensive care medicine.
Under normal conditions, lactate metabolism produces a small amount of ATP. However, by accelerating the glucose metabolism process, significantly more ATP can be produced. Therefore, a clinical condition associated with increased glucose metabolism may lead to elevated lactate levels, as the capacity of the Krebs cycle is limited. Many conditions and treatments in critically ill patients lead to an increase in glucose metabolism. One example is the increased sympathetic nervous system activation, which is markedly present in shock conditions. In such cases, lactate can serve as an exchangeable fuel between tissues (liver, kidneys, muscles) and even between cells (astrocytes, neurons) through lactate shuttles.
Unlike the oxygen-dependent Cori cycle (hepatic and renal gluconeogenesis), lactate metabolism allows for rapid energy production, and even exogenous lactate can be utilized as a fuel in this process. As a result, the traditional concept of lactate as an indicator of tissue hypoxia in shock conditions is being questioned. This relationship can be more problematic than suggested in guidelines, particularly in sepsis, where lactate clearance is impaired.
This review written on lactate may shed light on the management of critically ill children and may support the trend in the areas of research it suggests.
Response 1: Thank you for your thoughtful feedback and the time spent reviewing our work. We appreciate your insights and are glad the review has contributed to the ongoing discussion around lactate's role in critically ill pediatric patients.
Round 2
Reviewer 1 Report
Comments and Suggestions for Authors
The revised version looks acceptable to me. There are two remaining issues that I still have with the revised version, one major [1], one minor [2].
[1] The statement “ … from proton release during adenosine triphosphate (ATP) hydrolysis in glycolysis [20].” in line 155 is not correct.
The ultimate source of this incorrect claim that the source of systemic acidosis during lactic acidosis is ATP and not lactic acid, an investigator named Robergs, who has repeatedly published this biochemically incorrect claim. Several investigators have written comments etc. that explain the fallacy of Robergs’ claim that ATP hydrolysis is the source of systemic acidosis. Unfortunately this mistaken view still pops up regularly in the literature.
ADP is continuously regenerated to ATP in the cell, whereas the lactate anion leaves the cell together with a proton through an MCT transporter.
Stoichiometrically the correct net reaction is
Glucose -> 2 La- + 2 H+
The fact that in animals and patients acidosis and lactate levels are correlated but sometimes only poorly correlated (as you also refer to in your review) has to with mammalian acid-base physiology. H+ can be buffered by many systems including lungs and kidneys and the liver as well. These systems have different adaptation time scales (i.e. lung vs kidney) and may or may not function appropriately depending on the disease condition. See for example Rose’s textbook on acid-base physiology for an extensive explanation on buffering.
The reason I and many colleagues are so emphathic on this issue, is that this is not only a theoretical scientific debate, but also a clinical issue, which may lead to incorrect treatments.
So to summarize: Lactate production is coupled with acid production. Period.
In terms of revisions you only have to modify line 155 into for example:
“ … from coupled proton and lactate transmembrane transport during glycolysis [20].”
Leaving in ref [20] is not a problem.
[2]
When expressing lactate in mmol/L one decimal (e.g. 2.2 mmol/L) is usually used in the literature.
Author Response
Thank you for your feedback and thoughtful input. We understand your concerns regarding the statement in line 155 about proton release during ATP hydrolysis in glycolysis, and we appreciate the clarification. We have revised the sentence as you suggested.
As for the minor point regarding the expression of lactate in mmol/L, we have updated the values to one decimal place in line with standard practice in the literature. We believe these changes address the remaining issues and ensure the accuracy of the manuscript. Thank you again for your helpful comments!